# Development of a 3-D standardized lung template from low-dose CT scans*

Giulia Raffaella De Luca
*Department DEI*
*Alma Mater Studiorum - University of Bologna*
Cesena, Italy
giulia.deluca19@unibo.it

Mario Mascalchi
*Department SBSC*
*University of Florence*
Florence, Italy
mario.mascalchi@unifi.it

Stefano Diciotti
*Department DEI*
*Alma Mater Studiorum - University of Bologna*
Cesena, Italy
stefano.diciotti@unibo.it

*Abstract*—Lung imaging lacks a standardized reference space, hindering the large-scale, voxel-wise analyses that are routine in neuroimaging. To address this gap, we developed a high-resolution, open-source 3-D lung template and probabilistic lobar atlas from a cohort of 30 subjects from the National Lung Screening Trial (NLST). Created using a fully automatic pipeline based on the Advanced Normalization Tools (ANTs) ecosystem, this template reached convergence (dice similarity coefficient of 0.992 between consecutive iterations) after 11 iterations. We demonstrated its utility by registering 60 subjects with varying emphysema severity, finding that voxel-wise Jacobian analysis could distinguish disease-specific deformation patterns. This work provides a foundational, open resource for standardizing anatomical localization, enabling robust group-level studies in lung cancer screening research.

*Index Terms*—computed tomography, lung imaging, quantitative imaging, template.

## I. Introduction

Spatial normalization to a common reference frame, or template, is a cornerstone of modern neuroimaging, enabling voxel-wise group analyses and the creation of standardized coordinate systems for localizing anatomical and functional findings [1], [2]. This paradigm, however, is less established in lung imaging, where high inter-subject anatomical variability and respiratory motion pose significant challenges. The development of a standardized 3-D lung template, particularly from a large-scale lung cancer screening cohort, holds immense potential for advancing quantitative lung analysis [3]–[5]. A common reference space would also enable a standardized coordinate system for precise localization of lesions, e.g., nodules, facilitating consistent tracking and reporting across different time points and patients [6]. Furthermore, co-registering images to a standard template provides a harmonized input space for machine learning algorithms, improving model robustness and generalizability [7], [8] and disentangling anatomical variations from disease-induced ones [9], [10].

*The research leading to these results has received funding from the European Union - NextGenerationEU through the Italian Ministry of University and Research under PNRR - M4C2-I1.3 Project PE_00000019 "HEAL ITALIA" to Stefano Diciotti - CUP J33C22002920006. The views and opinions expressed are those of the authors only and do not necessarily reflect those of the European Union or the European Commission. Neither the European Union nor the European Commission can be held responsible for them.

Finally, such a framework is crucial for enabling population-level statistical analyses [6] and inter-subject comparisons of parenchymal characteristics associated with disease conditions such as pulmonary fibrosis [7], COPD, obesity, and cardiovascular calcifications [5].

## II. Related work

Previous efforts to create lung atlases have laid important groundwork, though many existing resources have limitations hindering widespread adoption. For instance, the work by Li et al. produced a valuable atlas but relied on manually selected anatomical landmarks, a process that limits scalability and reproducibility [3], [4]. Recently, Ryan et al. developed an open-source template-building pipeline based on the Symmetric Normalization (SyN) registration algorithm, but their reliance on binarized lung masks is a critical drawback [7]. Image intensity gradients drive the SyN algorithm to find an optimal transformation [11], [12], therefore using binary masks eliminates all internal anatomical texture, leaving only information at the lung boundary. This lack of a rich gradient field can lead to inaccurate alignments or non-physical internal deformations [13]. Other notable templates, such as those developed by Xu et al. and the one used in the ALIAS framework by Chen et al., are based on closed-source software or proprietary data, restricting transparency, external validation, and community-driven improvement [5], [6].

A common methodological choice in some of these works is the selection of an arbitrary "healthy" subject as the initial reference for template construction, which may introduce an uncharacterized bias into the final average-space template [6], [7]. Our work addresses these gaps by developing a 3-D lung template and lobar atlas using a fully automatic, open-source pipeline based on the widely-used Advanced Normalization Tools (ANTs) ecosystem [14]. The template is built from a representative, publicly available cohort from the National Lung Screening Trial (NLST), and we demonstrate its utility through a quantitative and voxel-wise analysis of deformations associated with pulmonary emphysema.

## III. Materials and methods

### A. Dataset description

We accessed de-identified data from NLST database under an approved data use agreement (CDAS Project Number: NLST-1175). All analyses complied with HIPAA regulations.

The NLST enrolled 53,454 current and former smokers (if they quit smoking within 15 years), aged 55-74 with at least a 30 pack-year smoking history. 26,722 participants were randomized to the low-dose chest CT (LDCT) arm and 26,732 to the control arm, which received posteroanterior chest radiography. Participants in the LDCT arm were invited to undergo three annual LDCT screening rounds, labelled T0 (baseline), T1, and T2. Active enrollment and screening were conducted between 2002 and 2007 across 33 participating centers in the United States [15].

To build the template, we initially selected 38 representative baseline LDCTs from unique NLST participants. This cohort had a mean age of 64.71 years (standard deviation, SD: 5.4 years), included 18 (47.37%) male participants, had a mean BMI of 28.26 kg/m² (SD: 6.52 kg/m²) and a mean smoking history of 64.64 (SD: 29.4) pack-years. Twenty-two participants (57.89%) had reported respiratory comorbidities. The scans were acquired from four different manufacturers (GE, Siemens, Philips, Toshiba) and reconstructed using four kernel types, with a mean slice thickness of 1.54 mm (SD: 0.37 mm). To mitigate bias from extreme anatomical variations and prevent cropping artifacts, an issue we identified in initial tests using ANTs default center of mass alignment, we refined our cohort based on the lung field-of-view (FOV). We calculated the FOV for all 38 LDCTs and included only the 30 ones between the 10th and 90th percentiles in the final template construction. The lungs and lobes sizes in the final cohort are presented as average (SD). Total lung volume is 5373.42 (905.27) cm³. The average left lung volume is 2491.87 (491.29) cm³, with average length 26.39 (1.85) cm, depth 19.56 (1.86) cm and width 14.10 (1.14) cm; the average right lung volume is on average 2881.55 (458.49) cm³, with average length 26.52 (1.79) cm, depth 19.98 (1.94) cm and width 15.21 (1.28) cm. The average lobes volumes are 1214.36 (265.94) cm³ for the left lower lobe (LL), 1277.51 (280.78) cm³ for the left upper (LU), 1289.33 (223.48) cm³ for the right lower (RL), 444.62 cm³ (128.31) for the right middle (RM) and 1147.59 (225) cm³ for the right upper (RU). To demonstrate the template's utility, we selected 60 LDCTs from unique participants for our emphysema application, divided into a severe emphysema group, a typical emphysema group, and a control group, each comprising 20 subjects. The demographic and clinical characteristics of these groups are summarized in Table I.

### B. Lung CT image processing

In a process analogous to neuroimaging skull-stripping [1], we first extracted the lungs from the chest LDCTs. Lungs were segmented using lungmask (v0.2.13), an open-source, pre-trained U-Net model for lungs and lobes segmentation

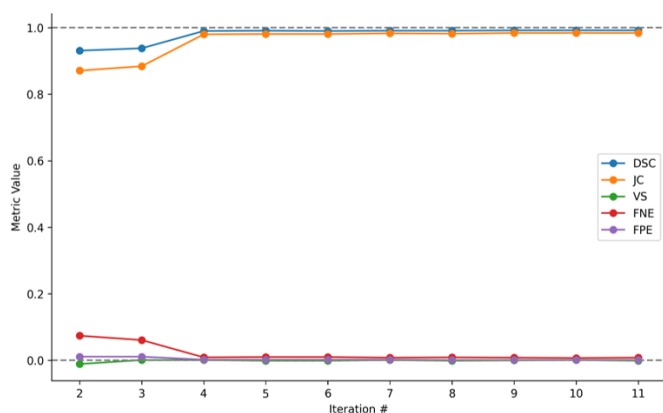

Fig. 1. Convergence of label overlap metrics during template construction. The plot shows the dice similarity coefficient (DSC), Jaccard coefficient (JC), volume similarity (VS), false negative error (FNE), and false positive error (FPE) calculated between binarized templates at consecutive iterations. DSC and JC range from 0 (no overlap) to 1 (perfect overlap); using this VS implementation, FNE and FPE indicate better alignment the closer they are to 0. Convergence was formally defined as a stable DSC > 0.90 with change < 0.001 for three consecutive iterations, a strict criterion that was met at iteration#11.

chosen for its state-of-the-art performance and robustness to pathological variations [16], [17]. Starting from these binary masks (with a value of 1 inside the lungs and 0 outside), we used the `ImageMath` tool from the ANTs toolkit to create a masked image where voxels within the lung retained their original Hounsfield Unit (HU) values, while all voxels outside the lungs were set to -2000 HU – a value well outside the physiological range to ensure only lung anatomy drives the registration process and facilitate template binarization, necessary to compute the similarity metrics during template building. We will refer to these images as masked LDCTs. The 3-D lung template was created using the diffeomorphic averaging framework implemented in the `antsMultivariateTemplateConstruction2.sh` script from the ANTs toolkit (v2.5.4) [12]. We designated

TABLE I
DEMOGRAPHIC AND CLINICAL CHARACTERISTICS OF THE EMPHYSEMA COHORT

| Characteristic | Control (N=20) | Typical (N=20) | Severe (N=20) |
|---|---|---|---|
| Age (years) | 58.95 (3.46) | 64.10 (4.88) | 64.25 (5.44) |
| Sex (Male) | 3 (15.0) | 12 (60.0) | 17 (85.0) |
| Race (White)[a] | 17 (85.0) | 20 (100.0) | 20 (100.0) |
| BMI (kg/m$^2$) | 26.97 (3.93) | 27.69 (7.62) | 25.24 (4.51) |
| Current Smoker | 16 (80.0) | 10 (50.0) | 0 (0.0) |
| Pack-years | 48.48 (21.53) | 57.28 (19.91) | 61.86 (24.42) |
| Comorbidities[b] | 7 (35.0) | 20 (100.0) | 20 (100.0) |
| %LAA (%)[c] | 0.0 (0.0) | 9.0 (0.0) | 44.0 (6.0) |

[*] Data are presented as mean (standard deviation) for continuous variables and N (%) for categorical variables.
[a] The cohort was predominantly White; other racial categories represented < 10% in any group.
[b] We considered respiratory related comorbidities.
[c] %LAA: Percentage of Low Attenuation Area (< -950 HU).

the masked LDCT of the participant with the largest lung FOV as the initial reference space, $T_0$, to prevent anatomical cropping during registration. The template construction process operates through iterative refinement, with each co-registration step employing the SyN algorithm [14],which has a long track record of success in the neuroimaging domain [18], [19], but especially for its top-ranking performance in the EMPIRE10 thoracic CT registration challenge, proving its suitability for this specific anatomy [7], [20]. While contemporary deep learning methods offer computational efficiency, their generalizability across unseen datasets can be a concern [21]. Given our primary goal of creating a robust and widely applicable template, the proven reproducibility and performance of SyN, within the ANTs C++ ecosystem, made it the most suitable choice. At each iteration, all the images undergo multi-stage registration to the current template: first, rigid transformation corrects for positional and orientational differences; next, affine transformation accounts for global size and shearing variations; finally, deformable SyN transformation performs local, non-linear warping to match the template's anatomy with high fidelity. This multi-stage process operates across a four-level multi-resolution pyramid (from coarse to fine) to capture transformations at different granularities. The resulting transformations and warped images are then averaged to obtain an average transformation and average shape, respectively. The updated template is then created by applying the inverse of the average transformation to the average shape. The parameters for the multi-resolution pairwise registration were set as follows. The maximum number of iterations at each resolution level (flag `-q`) was set to `100x70x50x10`, with fewer iterations at finer resolutions to account for the higher computational cost per iteration. The shrink factors (flag `-f`), which are the integer factors for downsampling the template image during registration, were set to `8x4x2x1`. Finally, the smoothing kernels (flag `-s`), specifying the standard deviation of a Gaussian smoothing kernel applied to the images before downsampling at each level, were set to `3x2x1x0` voxels. Rather than relying on a fixed number of build iterations, which defaults to four in the `antsMultivariateTemplateConstruction2.sh`, we implemented a quantitative convergence criterion to ensure the template reached maximum stability. To accommodate this, the script's maximum iteration parameter, $N_{max}$, was set to a conservatively high value of 15 that would not interfere with our stability-based termination. Building on Ryan et al.'s proposal, we assessed convergence at each iteration by computing the dice similarity coefficient (DSC) between the binarized templates of consecutive iterations ($T_i$ and $T_{i-1}$) using `LabelOverlapMeasuresImageFilter` from the Simple Insight Toolkit (SITK) [22]. Convergence was defined as DSC > 0.90 with the value remaining stable (< 0.001 change) for three consecutive iterations. The full iterative process is detailed in Algorithm 1.

---

**Algorithm 1** Iterative Template Creation

---
1: **Initialize:** Template $T_0$ from subject with max FOV; subject images $\{S_k\}_{k=1}^{29}$.
2: **for** iteration $i = 1$ to $N_{\max}$ **do**
3:    *// Register each subject to current template*
4:    **for** each subject $S_k$ **do**
5:       **for** each resolution level **do**
6:          Compute transformation $\phi_{k,i}$ mapping $S_k \to T_{i-1}$ using a multi-stage registration (Rigid $\to$ Affine $\to$ SyN).
7:       **end for**
8:    **end for**
9:    *// Create new template from averaged transformations*
10:    Compute the average transformation $\bar{\phi}_i$ from the set $\{\phi_{k,i}\}$.
11:    Compute the average shape $A_i$ by warping and averaging all subjects into the common space: $A_i = \frac{1}{N}\sum_{k=1}^{N} \phi_{k,i}(S_k)$.
12:    Update the template by applying the inverse of the average transformation to the average shape: $T_i = \bar{\phi}_i^{-1}(A_i)$.
13:    *// Check convergence*
14:    Binarize $T_i$.
15:    Compute similarity metrics: DSC, Jaccard, FNE, FPE, VS.
16:    **if** DSC > 0.90 and $\Delta$DSC < 0.001 for 3 consecutive iterations **then**
17:       **CONVERGED**
18:    **end if**
19: **end for**

---

### C. Consensus lobar atlas

For the 30 subjects in the template cohort, the five lobes were segmented in their native space using *lungmask* [16]. The final individual transformations generated during the lung template construction were applied to these native-space lobar masks to warp them into the final template space using nearest-neighbor interpolation, with the `antsApplyTransform` script. These 30 warped masks for each lobe were then averaged on a voxel-wise basis to create a probabilistic map, where each voxel's value represents the probability of it belonging to that lobe. A deterministic atlas was also created by assigning each voxel to the lobe with the highest probability, using the `ImageMath MostLikely` function, part of the ANTs package, with a minimum probability threshold of 0.4.

### D. Emphysema application

To demonstrate the template's utility, we performed a voxel-wise analysis of emphysema-related structural changes. This involved identifying distinct groups from the NLST cohort, co-registering them to our template, and analyzing the resulting spatial patterns of lung tissue deformation. Building on previous work on emphysema evaluation in NLST [24], we quantified emphysema as the percentage of low attenuation area (%LAA), defined as the percentage of lung volume with voxel value below –950 HU. From a subset of 4,952

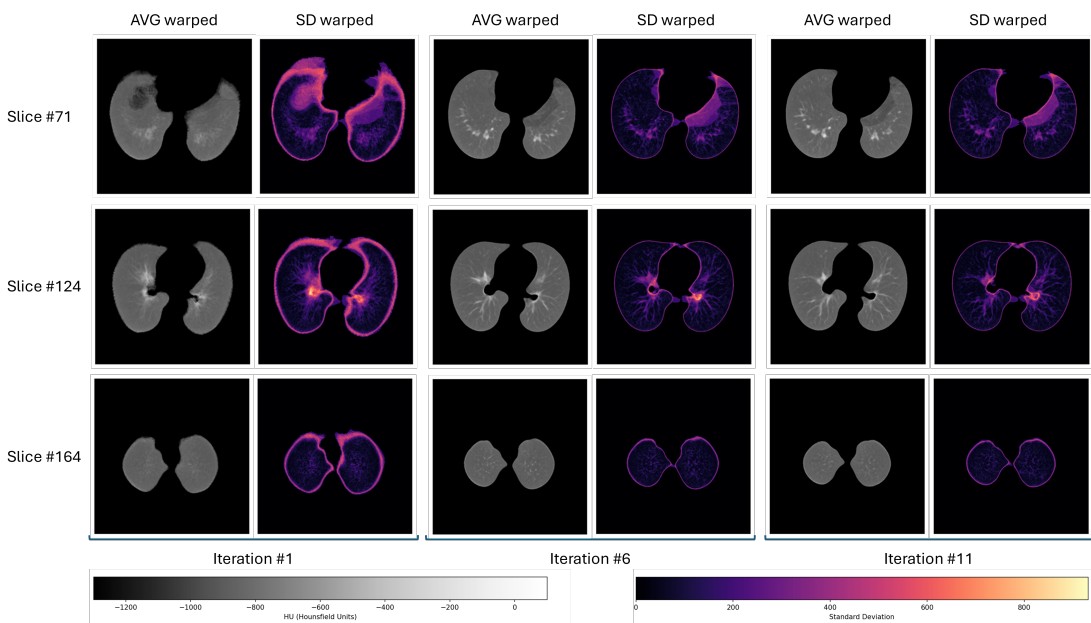

Fig. 2. Axial view of average (AVG) and standard deviation (SD) across warped LDCTs for iterations # 1, #6 and #11 (from left to right), in slices #71, #124, and #164 (from top to bottom). All figures of the lung are in "radiological" convention, where the left side of the image is the right lung.

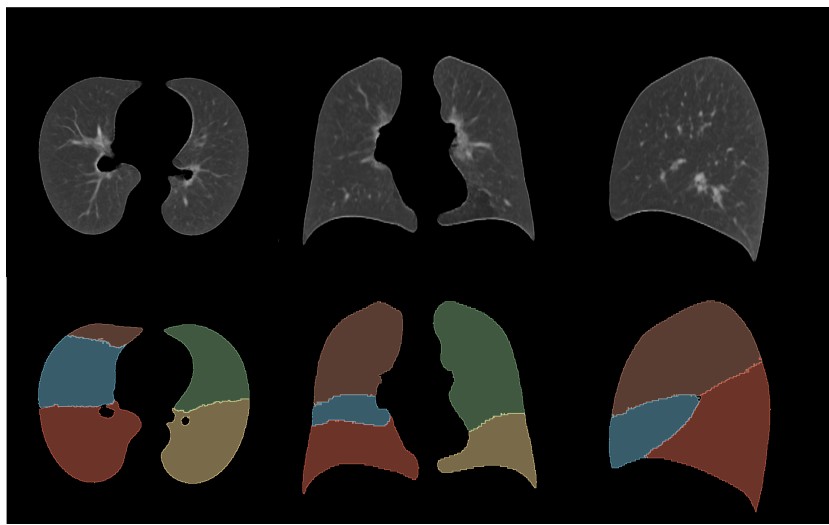

Fig. 3. LDCT lung template (top panel) and corresponding deterministic lobe maps in the template space (bottom panel) in the axial, coronal, and sagittal planes (left to right). Lobe color coding: right lower lobe (red), right middle lobe (blue), right upper lobe (brown), left upper lobe (green), left lower lobe (yellow). All figures of the lungs are in radiological convention, where the left side of the image is the right lung.

participants in the NLST CT arm (Ann Arbor cohort), we first extracted a high-quality sample of baseline examinations suitable for emphysema quantification. This "quantification sample" included LDCTs that met two quality criteria: a slice thickness $\leq$ 2.5 mm and the use of a soft or mid-soft reconstruction kernel [25]. When multiple scans were available for a participant, we prioritized those meeting both criteria. For each scan in this sample, we created a binary %LAA mask (1 where HU $< -950$, 0 otherwise) by first isolating the lungs using lungmask [16], and then thresholding the voxels at –950 HU and determined %LAA. Using these masks and available clinical data, we formed three distinct study groups (n=20 each):

- Severe emphysema group: LDCTs associated to participants with a clinical diagnosis of emphysema (as reported in NLST database) and the highest %LAA in the quantification sample.
- Typical emphysema group: LDCTs associated to participants with a clinical diagnosis of emphysema whose %LAA values were closest to the median %LAA in the quantification sample. This group is intended to represents typical, smoking-related centrilobular emphysema.

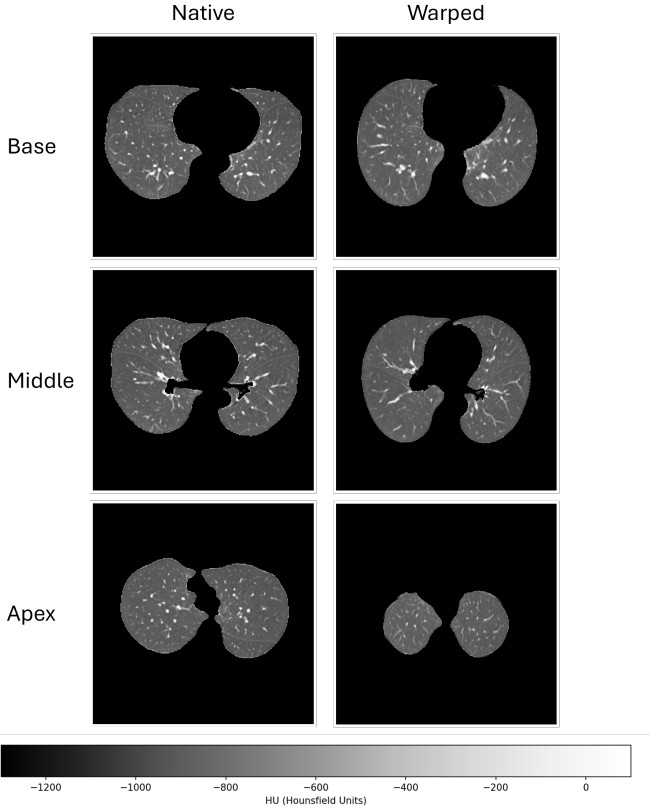

Native    Warped

Base

Middle

Apex

HU (Hounsfield Units)

Fig. 4. Axial views of one sample from the template cohort before and after warping in the template space, in base (35%), middle (50%) and apex (80%) height (from top to bottom). All figures of the lung are in "radiological" convention, where the left side of the image is the right lung.

- Control group: LDCTs associated to participants with no clinical diagnosis of emphysema and the lowest %LAA values in the quantification sample. This "extreme-healthy" selection was a deliberate choice to establish a clear, low-noise baseline against which disease patterns could be robustly compared for this preliminary utility study.

Each of the 60 selected LDCTs was registered to our final lung template using the same registration parameters employed during template construction with `antsRegistration` script. The resulting deformation fields were then applied to each corresponding binary %LAA mask using nearest-neighbor interpolation with `antsApplyTransform` script, mapping the spatial distribution of emphysemateous regions into the common template space. To quantify local anatomical changes, we calculated the Jacobian determinant of each deformation field using ANTs `CreateJacobianDeterminantImage.sh` script. The Jacobian determinant measures local volume change at each voxel resulting from the registration; values greater than 1 indicate expansion, values less than 1 indicate tissue contraction relative to the template [26]. We then performed a group-wise comparison by calculating the mean and standard deviation of the Jacobian determinant maps for each group to identify regions of structural divergence.

## IV. RESULTS

### A. 3-D lung template

The lung template generation process converged after 11 iterations, meeting our pre-defined stability criterion with a final average DSC of 0.992 (Fig. 1). The entire process took 117 hours on a 228-core Linux server. The final template has dimensions of 512×512×205 voxels. Figure 2 qualitatively illustrates the refinement of the template, showing a progressive sharpening of anatomical features and a reduction in the standard deviation across the population. The final template has a total lung volume of 5435.14 cm³. The left lung volume is 2525.33 cm³ (length 25.5 cm, depth 19.26 cm, width 13.72) and right lung volume is 2909.82 cm³ (length 26.18 cm, depth 19.51 cm, width 14.3 cm). The resulting probabilistic lobar atlas provides detailed spatial distributions for each lobe (Fig. 3) with mean volumes of LL: 1220.81, LU: 1304.52, RL: 1295.97, RM: 445.29, and RU: 1168.55 cm³. These three-dimensional measures of the final lung template are both consistent with the average ones from the 30 subject cohort in the native space and the healthy lung template that Ryan et al obtained from 62 COPDGene participants [7]. Figure 4 shows the result of anatomical normalization achieved by warping an image in the template space. The 3D lung template and probabilistic lobar atlas are publicly available in the Zenodo repository (https://doi.org/10.5281/zenodo.17159622) under a CC-BY 4.0 license. Resources are provided in NIfTI format with a companion Jupyter notebook for interactive visualization.

### B. Emphysema-related deformation

Warping the selected LDCTs from the quantification sample to the template space enabled a group-level analysis of emphysema-related deformation changes. On our server, this process required approximately 20 minutes per image for the registration step and a few seconds for applying the transformation. The mean Jacobian determinant maps (Fig. 5, top row) revealed spatially consistent patterns of deformation when comparing the emphysema groups to the control group. Specifically, the difference between the mean Jacobian maps highlights regions of relative expansion (Jacobian > 1, purple and pink areas) in the emphysema groups, while contraction (Jacobian < 1, black areas) can be observed in the healthy group. In the severe group, expansion occurs especially at the lung bases, presumably reflecting compensatory hyperinflation of diaphragm in presence of severe emphysema, while in the typical group it is spread all over the lung, and is generally less pronounced. Furthermore, the analysis of the standard deviation (Fig. 5, bottom row) reveals higher deformation heterogeneity in the emphysema groups compared to the control. The higher variability in the emphysema groups demonstrates that the structural changes induced by the disease are highly variable in their spatial pattern and severity, adding a significant layer of complexity on top of the normal anatomical differences found within the control population.

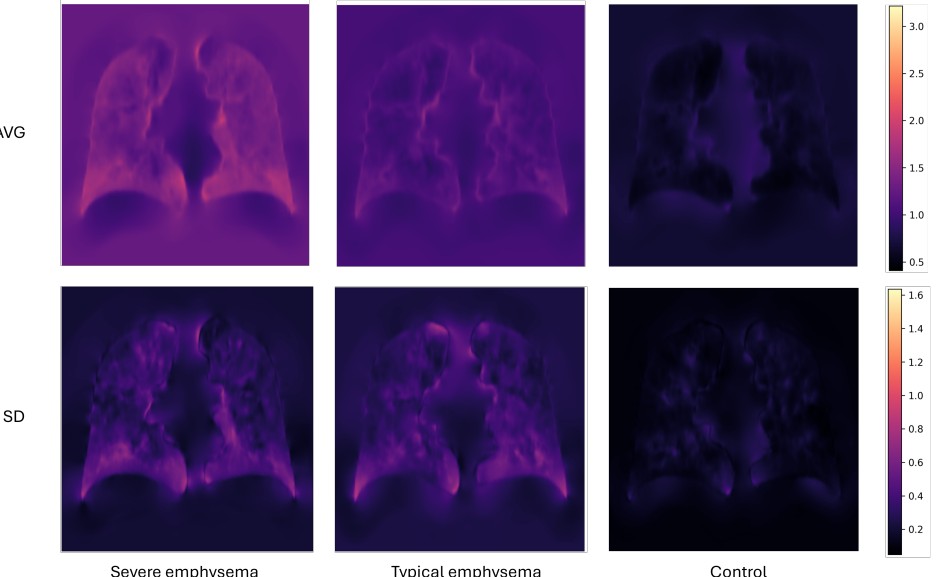

Fig. 5. Coronal views of the Jacobian determinant analysis across the three emphysema groups. Top row): Average (AVG) Jacobian determinant maps for the severe emphysema (left), typical emphysema (center), and control (right) groups. Values > 1 indicate local expansion was required to map the subject to the template space. Bottom row: Corresponding standard deviation (SD) maps for each group. All lung images follow the radiological convention, with the left side of the image corresponding to the right lung.

## V. DISCUSSION

We have developed and applied a high-resolution, open-source LDCT 3-D lung template and a probabilistic lobar atlas from the NLST cohort. Our work provides a foundational resource for the lung imaging community by combining several methodological choices to ensure robustness and reproducibility. By employing a fully automated pipeline built on the open-source ANTs ecosystem, we ensure transparency and avoid the challenges of proprietary software or laborious manual landmark selection. Furthermore, our use of masked images where the lungs maintain original Hounsfield Unit intensity, allows registration to leverage rich parenchymal texture, a key principle for achieving physically plausible alignments. Finally, we mitigate a potential source of bias by selecting the initial reference for template construction based on an objective criterion (largest FOV) rather than an arbitrary subject. These combined choices distinguish our approach and are critical for creating a robust reference space for standardized analyses.

The successful construction of a sharp, low-variance template from a diverse set of multi-vendor, multi-kernel scans demonstrates the robustness of the ANTs-based registration framework. The final template's dimensions and volumetric asymmetry between the right and left lungs are consistent with known human anatomy, providing face validity for the averaging process. Our emphysema application study underscores the template's practical utility. By warping subjects into this common space, we were able to link a densitometric biomarker (%LAA) to a morphometric one (Jacobian determinant). The mean Jacobian maps successfully identified group-level structural alterations linked to emphysema and even

revealed distinct patterns between typical and severe disease, demonstrating a capacity for patient stratification. Simultaneously, the standard deviation analysis provided a quantitative measure of variability, allowing us to distinguish the consistent signature of normal anatomical variation in controls from the highly heterogeneous expression of the disease in patients. This ability to spatially resolve and quantify known disease patterns validates our framework's anatomical correctness.

Our study has several limitations. The first set of limitations relates to the template's construction. The cohort of 30 subjects, while intentionally diverse—including scans from four manufacturers and with four reconstruction kernels—is small and may not capture the full variability of the larger NLST population. The specific impact of radiological factors like reconstruction kernel choice on the final template morphology has not yet been quantified and remains a key area for future work. Additionally, our initialization strategy, while pragmatically chosen to prevent the anatomical cropping observed in initial tests, could introduce a potential bias that requires future systematic investigation. The second set of limitations pertains to our study's validation. Critically, template convergence was evaluated using DSC. We acknowledge that this global overlap metric does not guarantee the precise alignment of fine anatomical landmarks. Validation on expert-annotated images will not only provide a truer measure of anatomical fidelity but also enable the development of more sophisticated, context-aware stopping criteria, potentially reducing the number of iterations needed for convergence. Regarding the analysis on emphysema-related deformation, our validation cohorts were selected based on representative (median) and extreme (lowest) %LAA values to maximize contrast, rather than

being explicitly matched for all potential confounders, which could influence the Jacobian analysis. Voxel-wise statistical analysis with threshold free cluster enhancement to formally test for group differences and correct for multiple comparisons will be performed on the subsequent template derived from a larger cohort. Perhaps the most important consideration is the template's inherent scope. The template is derived exclusively from the NLST cohort, which consists of 55-74 years old current and former heavy smokers. Consequently, it represents the anatomy of this specific population, not a general-purpose "healthy human lung". Given that long-term smoking can induce subtle parenchymal and airway changes, caution is warranted when applying this template to other populations. Therefore, while this "NLST-space" template is an ideal reference for studies within NLST or similar lung cancer screening cohorts, our long-term vision is to develop a library of population-specific atlases to address the distinct anatomical characteristics of different groups (e.g., stratified by sex, age, BMI, or disease status).

This publicly available template and atlas can serve as a standard reference space for lung imaging, analogous to the MNI152 template in neuroimaging. This will support more reproducible nodule localization (e.g., defining a standardized (x,y,z) coordinate system and reporting nodule's location in this common "NLST-space"), facilitate multi-site data aggregation, and provide a harmonized input for developing and validating AI models. Future work will focus on expanding the template cohort to create population-specific atlases (e.g., stratified by sex, BMI, or smoking status). A critical next step is to use the generated transformations to warp the full, non-masked CT images into the template space to create a population-level atlas of lung density and pathology, including the spatial distribution of lung nodules.

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
