# OpenReview forum: "Development of a 3-D standardized lung template from low-dose CT scans"
_IEEE.org/EMBS/BHI/2025/Conference — BHI 2025_

### Official Review · Reviewer_nyEb · 2025-07-12
**Development of a 3-D standardized lung template from low-dose CT scans**

**Confidence:** 3
**Clarity Of Writing:** good
**Clinical Significance:** good
**Methodological Novelty:** good
**Overall Rating:** 5
**Final Rating:** 7

**Experiments And Results:**

fair

**Questions For The Authors:**

1. You used lungmask (v0.2.13), a pre-trained U-Net model, for lung and lobe segmentation. Could you clarify why this specific model was selected? Was it evaluated against more recent or specialized models?

2.Why ANTs was chosen exclusively over other robust registration frameworks like Elastix or Greedy?

3. One key limitation is the absence of performance comparison with existing lung template or segmentation frameworks. Specifically:

    a. How does your pipeline's Dice, Hausdorff distance, runtime, or memory efficiency compare with other state-of-the-art methods like CT-LungNet, VoxelMorph, or lungCT?

    b. Was there any measurable improvement in registration accuracy, segmentation robustness, or anatomical plausibility over these existing approaches?


4. In Figure 1, the label overlap metrics (e.g., Dice, Jaccard) appear to converge by iteration 4-5. What was the reasoning for continuing the template construction for 11 iterations?


5. How does your proposed pipeline perform relative to execution time, and robustness?

6. Have you considered adding 3D contour plots or other interactive 3D visualizations to improve the interpretability of the lung template and atlas results?

7. Did you compute and compare mean, standard deviation, and other descriptive statistics (e.g., volume, surface area, width, depth, height) for each lung lobe (LUL, LLL, RUL, RML, RLL)?

    a. Were these compared with a ground truth or reference standard?

    b. Did you perform any statistical significance testing (e.g., t-tests, confidence intervals) to validate those differences?

**Strengths:**

The paper is clearly written, with well-described methods and effectively presented results. Including a graphical representation of the methodological steps would further enhance clarity and accessibility.

**Summary Of The Paper:**

The authors present a high-resolution, open-source 3D lung template along with a probabilistic lobar atlas, developed using a fully automated pipeline built on the Advanced Normalization Tools (ANTs) framework.

**Weaknesses:**

The contribution of the paper is moderate, as it primarily involves building an image processing pipeline for a well-defined task using existing tools. While the authors utilized Advanced Normalization Tools (ANTs), the rationale for exclusively choosing ANTs over other established alternatives—such as Elastix, VoxelMorph, deep learning-based methods like CT-LungNet, or R-based solutions like the lungCT package—is not adequately discussed. Additionally, the paper lacks comparative evaluation with these existing methods, which limits the assessment of its relative effectiveness and novelty.

---

### Official Review · Reviewer_g2tc · 2025-07-15
**This paper presents a well-executed technical contribution developing an open-source lung template from NLST data, though it suffers from limited dataset.**

**Confidence:** 3
**Clarity Of Writing:** good
**Clinical Significance:** great
**Methodological Novelty:** fair
**Overall Rating:** 6
**Final Rating:** 7

**Experiments And Results:**

fair

**Questions For The Authors:**

1. caption 5: 'follo'
2. How sensitive is the final template to the choice of initial reference subject, and have you tested alternative initialization strategies?
3. Have you tested the template's performance with different scanning protocols, reconstruction parameters, or scanner manufacturers beyond those represented in NLST?
4.  Can you provide more detailed analysis of the computational requirements and processing times for end-users who might want to register their data to this template?
5. Do you plan to expand the template to larger cohorts?

**Strengths:**

The paper addresses a genuine need in lung imaging for standardized reference spaces analogous to neuroimaging templates. The methodological approach is well-designed, using masked images that preserve internal texture while eliminating non-lung anatomy, which is superior to binary mask approaches used in previous work. The fully automated pipeline based on the widely-used ANTs ecosystem ensures reproducibility and transparency. The convergence criteria are clearly defined and rigorously applied, with comprehensive similarity metrics beyond just DSC. The validation through emphysema analysis provides concrete evidence of the template's utility for detecting disease-specific patterns. The paper includes appropriate technical details about registration parameters and computational requirements. The commitment to open-source release represents a valuable contribution to the research community. The demographic and clinical characterization of the cohorts is thorough and appropriate for the intended application.

**Summary Of The Paper:**

This paper develops a high-resolution 3D lung template and probabilistic lobar atlas from 30 subjects in the National Lung Screening Trial  using a fully automatic pipeline based on Advanced Normalization Tools. The template creation process employs masked CT images, which preserve internal Hounsfield Unit values, and uses iterative diffeomorphic averaging with multi-stage registration (rigid, affine, deformable SyN). Convergence was achieved after 11 iterations with a dice similarity coefficient of 0.992. The paper demonstrates usefulness through voxel-wise Jacobian analysis showing that the template can identify disease-specific deformation patterns.

**Weaknesses:**

1. lacks comparison with other existing lung templates or atlases to establish relative performance or advantages.
2. Statistical significance testing is absent from the validation analysis
3. A brief guidance on when this template should or shouldn't be used for different populations or applications could be included.
4. caption of Fig. 5: 'follo'
5. As all subjects are smokers aged 55–74, the template may not generalize to healthy, younger, or non-smoking populations.
6. The paper argues that this template could enhance AI model robustness, but no actual machine learning application is demonstrated.

---

### Official Review · Reviewer_V6BL · 2025-07-15

**Confidence:** 4
**Clarity Of Writing:** good
**Clinical Significance:** good
**Methodological Novelty:** fair
**Overall Rating:** 6

**Experiments And Results:**

good

**Questions For The Authors:**

.

**Strengths:**

The authors propose an automated and open-source pipeline using ANTs and lungmask (a U-Net model), thus avoiding manual landmarking. The paper is well-written, logically structured, and the methods are described in detail.

**Summary Of The Paper:**

The authors describe the development of a high-resolution, open-source 3D lung template and probabilistic lobar atlas derived from NLST. Using the ANTs ecosystem, the authors propose a fully automated pipeline for iterative construction of the template. Further, they register 60 LDCTs stratified into various ephysema stages and use it to reveal disease specific deformation patterns. This work is an important step towards facilitating a freely available reference space for lung imaging, analogous to MNI152 in neuroimaging.

**Weaknesses:**

1. Figure 5 visually contrasts deformation fields between severe emphysema, typical emphysema, and control. While the qualitative description is very helpful but quantitative analysis is required to judge whether these differences are meaningful or within group variability. This is important as the emphysema application is central to this paper.
2. It would also be very helpful if the authors can discuss the anatomical accuracy  - landmarks of the lung align meaningfully. DSC only measures the global mask overlap that is insufficient especially in the case of medical imaging.
3. Algorithm 1’s pseudocode refers to “for iteration i=1 to Nmax” but Nmax is undefined; please specify Nmax

---

### Official Review · Reviewer_EAKk · 2025-07-16
**A valuable study with high potential being used in practical**

**Confidence:** 3
**Clarity Of Writing:** good
**Clinical Significance:** good
**Methodological Novelty:** good
**Overall Rating:** 7
**Final Rating:** 7

**Experiments And Results:**

great

**Questions For The Authors:**

Could you show the original CT image along with the template and the mask in Figure 2.? It's good to see how the templates and masks iterate but it's also important to know how well the template mapping doing with input CT images.

**Strengths:**

The authors provided a well-optimized pipeline to build lung template and lobar atlas with low-dose chest CTs.

The author made the whole work open-source, which can benefit the community and be combine with AI techniques in the future.

The method showed good robustness when applied on multi-vendor multi-kernel scans.

The method was tested on lung disease (emphysema) diagnosis with detailed analysis, proving its usability in practical.

**Summary Of The Paper:**

The authors developed an open-source lung template and a probabilistic lobar atlas using low-dose chest CTs from 30 subjects, and tested it on subjects with different emphysema severities.

**Weaknesses:**

As summarized in the discussion section, the dataset used for developing the pipeline was small.

The layout can be optimized. The Figure 1. is far from the corresponding text content. The connection between paragraphs also can be refined. There are some minor typos, for example, the word "follo" in the caption of Figure 5.

---

### Official Review · Reviewer_2FNd · 2025-07-18
**Population-Specific Lung Template**

**Confidence:** 3
**Clarity Of Writing:** great
**Clinical Significance:** good
**Methodological Novelty:** good
**Overall Rating:** 7

**Experiments And Results:**

good

**Questions For The Authors:**

1. Do you have plans to validate the template on external datasets from different institutions or imaging protocols?
2. How would the template perform when applied to non-smokers, younger populations, or patients with different lung pathologies?

**Strengths:**

1. The authors developed an open-source and automated pipeline that is publicly available.
2. The work addresses a fundamental gap in lung imaging by creating the first standardized reference space.
3. The authors demonstrated robust convergence with comprehensive validation metrics, achieving high similarity scores (DSC=0.992) across multiple measures including Jaccard coefficient and volume similarity.

**Summary Of The Paper:**

This paper develops a high-resolution, open-source 3D lung template and probabilistic lobar atlas from 30 NLST low-dose CT scans using an automated ANTs pipeline that converged after 11 iterations (DSC=0.992), and demonstrates its clinical utility through voxel-wise Jacobian analysis that successfully distinguishes emphysema severity patterns across 60 subjects.

**Weaknesses:**

1. The template construction was limited to only 30 subjects.
2. The cohort consists exclusively of heavy smokers aged 55-74 years.
3. The initialization strategy using the largest field-of-view subject could introduce systematic bias